# Experimental Study on Laser Ablation Texture-Assisted Grinding of Tungsten Alloy

**DOI:** 10.3390/ma15197028

**Published:** 2022-10-10

**Authors:** Bing Chen, Ye Guo, Shunshun Li, Guoyue Liu

**Affiliations:** 1Hunan Provincial Key Laboratory of High Efficiency and Precision Machining of Difficult-to-Cut Material, School of Mechanical Engineering, Hunan University of Science and Technology, Xiangtan 411201, China; 2Hunan Taijia New Material Technology Co., Ltd., Changsha 410203, China

**Keywords:** laser ablation texture, assisted grinding, grinding force, surface quality, tungsten alloy

## Abstract

In order to machine the tungsten alloy with high efficiency, low damage and precision, laser ablation texture technology and precision grinding technology were combined to carry out grinding experiments of tungsten alloy and laser ablation texture-assisted grinding experiments. The advantages of laser ablation texture-assisted grinding tungsten alloy were investigated by comparing of the surface morphology, grinding force and surface roughness between ordinary grinding and laser ablative texture-assisted grinding. The results demonstrate that the surface morphology of ordinary grinding tungsten alloy was poor, the surface roughness was relatively high and the grinding force was relatively large. The surface morphology of the laser ablation texture-assisted grinding tungsten alloy processed by laser ablation texture was improved, the surface roughness decreased by 0.023 µm–0.204 µm, the normal force decreased by 49.91–59.46% and the tangent force decreased by 44.11–58.49%. Meantime, for the area ratio of texture A being the most, the grinding effect was related to the area ratio of texture, and the lowest grinding force and the best surface quality were observed on the tungsten alloy with the laser ablated texture A; the grinding forces and roughness of the other textures’ workpiece was similar and close because of their similar area ratios. The results demonstrate that laser ablation texture-assisted grinding of tungsten alloy could improve machining quality and reduce grinding force, which would provide guidance for realizing the high efficiency and precision machining of tungsten alloy.

## 1. Introduction

Tungsten alloy is a kind of biphasic composite material composed of a tungsten phase and bonding phase. Many excellent physical and chemical properties are presented on tungsten alloy, such as high density, high strength (1000–1700 MPa), high ductility (10–30%) and strong corrosion resistance, etc. Therefore, it is widely used in civil and military industries [1]. At present, tungsten alloy machining methods mainly include cutting [2], grinding and polishing [3]. In the traditional cutting process, due to the large differences between the characteristics of biphase materials in tungsten alloy, the workpiece appears with scaly spines and furrows, and a series of problems such as tool wear, poor machining integrity and chip adhesion occur, which makes it difficult to obtain the high surface quality of tungsten alloy, and limits the application of tungsten alloy [4]. In view of the above problems in machining, some scholars proposed ultrasonic elliptical vibration machining technology [5] and low-temperature cutting technology [6], but the improvement of tungsten alloy machining quality by these technologies is limited.

Grinding technology is a wide range of applications, and is suitable for the precision machining of many kinds of difficult-to-cut materials. However, there are some problems such as large grinding force, poor grinding quality and high grinding temperature when machining tungsten alloy. Zhou et al. [7] carried out an experimental research by the electrolytic grinding of tungsten alloy, and the research demonstrated that different degrees of corrosion pits appeared on the surfaces of tungsten alloy after electrolytic grinding; besides, compared with mechanical grinding, the grinding force of the electrolytic grinding was reduced to only 1/3 to 1/2 of that of mechanical grinding, and the residual stress was also reduced. Xu et al. [8] studied the wear mechanism of diamond tools in the rotary ultrasonic grinding of tungsten alloy; it was demonstrated that diffusion wear was exhibited on diamond in the absence of oxygen, and diffusion wear and oxidation wear was exhibited on diamond in the condition of oxygen. In the meantime, the ultrasonic grinding process could not only reduce the grinding force and grinding temperature, but also reduce the wear area of diamond. Shi et al. [3] carried out an experimental study on the grinding and polishing of tungsten alloy and silicon carbide sandpaper, with a particle size of 38 μm, 25 μm, 18 μm and 15 μm, and a diamond consolidation abrasive grinding pad with a particle size of 8 μm was used for grinding. It was found that the particle size was smaller and the ground surface was smoother; meanwhile, it was demonstrated that the height difference between the tungsten phase and the bonding phase was obvious on the surface of the polished tungsten alloy when using a silica sol polishing solution with an average particle size of 72 nm, in which the tungsten was higher and the bond was lower. This was due to the different hardness of the tungsten phase and the bonding phase; in addition, the acidic environment could inhibit the pit phenomenon on the surface of tungsten alloy, and the surface quality was better. Although the grinding and polishing of tungsten alloy can improve the damage phenomenon of the previous process, there are still problems, such as the low processing efficiency and the height difference of the polishing surface, which are caused by the difference in characteristics of the two phases [9].

Laser-assisted grinding could reduce the grinding force and grinding temperature, and improve surface quality [10,11]. In the laser-assisted grinding process, the laser beam is focused on the surface of the workpiece to be processed, and the energy of the laser is converted into heat energy, which softens the workpiece material, thus reducing the shear strength of the workpiece and making the difficult material easier to be removed in the subsequent secondary processing [12,13]. At present, the laser-assisted machining has been applied to difficult-to-cut materials [14], such as alumina, silicon nitride [15] and titanium alloy [16], and so on. Azarhoushang et al. [17,18,19] carried out grinding experiments on silicon nitride materials with laser ablation texture and non-ablation texture on the surface, respectively, and the research demonstrated that laser ablation texture could sharply reduce the tangential grinding force, normal grinding force and surface roughness, and the reason of reducing the grinding force was the laser ablation texture, which would reduce the contact area between the cutting tool and workpiece machining surface, reducing the relative friction; at the same time, the ablation textures could store grinding fluid and abrasive dust [20], which reduced the grinding temperature. Zhang et al. [16] used a laser marking machine to ablate six different textures on the surface of titanium alloy, and carried out grinding comparative experiments on the titanium alloy workpiece; the results demonstrated that the grinding force of the laser-ablated texture workpiece was reduced by 45–56%, and the grinding temperature was reduced by 41–52% with the same machining conditions. The advantages of laser ablation texture-assisted grinding of titanium alloys were demonstrated. Kadivar et al. [21] used a nanosecond laser to ablate the grinding wheel and the silicon nitride workpiece, on which textures of a certain shape and size were ablated and carried out on micro-grinding experiments; the research results demonstrated that the grinding force was reduced by nearly 40% when the laser ablation area of the silicon nitride workpiece was 30%.

At present, there are a lot of studies on the cutting, grinding and polishing of tungsten alloys; however, there are many problems in traditional cutting and grinding. Meanwhile, the laser ablation texture grinding technology has been used to machine other difficult-to-cut materials with high efficiency and surface quality, while there are few reports on the high efficiency, precision grinding or laser ablation texture-assisted grinding of tungsten alloy. Therefore, experimental research on the laser ablation texture-assisted grinding of tungsten alloy was carried out in this paper. By comparing the surface morphology, grinding force and surface roughness of tungsten alloy machined by ordinary grinding and laser ablation texture-assisted grinding, the advantages of laser ablation texture-assisted grinding for tungsten alloy were discussed. This paper provides guidance for realizing the high efficiency and precision machining of tungsten alloy.

## 2. Components of Experimental Tungsten Alloy

The tungsten alloy material used in the experiment is 95W-3.5Ni-1.5Fe, and the size of the tungsten alloy workpiece is 20 mm × 10 mm × 10 mm. Tungsten alloy is a biphase composite material composed of a tungsten phase and bonding phase. The content of the tungsten phase is about 95%, and the main components of the bonding phase are Ni and Fe. This also avoids the bad characteristics of the high brittleness of pure tungsten, and significantly improves the corrosion resistance and machining performance [22,23]. The main components of tungsten alloy are shown in Table 1.

## 3. Experimental Scheme

In order to explore the superiority of the laser ablation texture-assisted grinding tungsten alloy, laser ablation texture-assisted grinding experiments were divided into two stages. In the first stage, a certain size and shape of the texture on tungsten alloy were ablated with a nanosecond laser; in the second stage, surfaces without ablation texture and with ablation texture on the tungsten alloy workpiece were ground. The grinding force data in the machining process was collected, and the surface morphology and roughness of the processed tungsten alloy were measured.

### 3.1. Experimental Scheme of Laser Ablation Texture

A nanosecond laser marking machine (Shenzhen, China, Fire Alarm Manufacturer Orena) was used to carry out laser ablation on the surface of tungsten alloy materials. The laser ablation system is mainly composed of the nanosecond laser source, tungsten alloy ablation materials and workbench, as shown in Figure 1a. According to the experience of scholars on the ablation texture of the refractory metal materials [19], the four textures [17,21] in Figure 1b were designed. The depth, width and spacing of the grooves in the four textures were all the same, which were 70 μm, 70 μm and 400 μm, respectively. Texture A was composed of many semicircles with aradius of 200 μm and 270 μm, texture B was a groove formed by two mutually perpendicular straight lines, where the angle α between the line and the horizontal direction is 45°, texture C was similar to the “W” shape, which was composed of straight lines, and texture D was different from texture B in that one of the lines coincides with the horizontal direction. According to the designed texture size, the area ratio occupied by the four textures could be calculated. The area ratio of texture A was 49.255%, the area ratio of texture B was 31.938%, the area ratio of texture C was 23.759%, and the area ratio of texture D was 31.938%. During the laser ablation texture experiment, the nanosecond laser marking machine first focused on the ablation surface of tungsten alloy, and then, based on the laser ablation parameters as shown in Table 2, four different textures were ablated on the surface of tungsten alloy by scanning the reciprocating Z-line. The depth of 70 μm could be reached after two times of test ablation, and the second ablation was performed directly without focusing. After ablation, the surface morphology of four textures was observed by electron microscopy and scanning electron microscopy (JSM-IT100, Tokyo, Japan, JEOL).

Figure 2 shows the textures (type A, type B, type C and type D) SEM morphology after nanosecond laser ablation on the tungsten alloy surfaces. As shown in Figure 2, the texture grooves on the surfaces of the tungsten alloy material were clearly visible, which agreed well with the designed theoretical image.

As shown in Figure 3, the width, spacing and depth of textures were observed by electron microscopy, the actual width of the grooves in texture A was 73.3260 µm and 76.3813 µm, the actual depth of the grooves was 70.2718 µm and 76.3465 µm, and the actual spacing of the grooves was 409.4419 µm and 418.5804 µm. The actual width of the grooves in texture C was 71.3255 µm and 73.2360 µm, the actual depth of the grooves was 70.2708 µm and 76.3896 µm, and the actual spacing of the grooves was 415.5589 µm and 418.6139 µm. The actual size was slightly different from the theoretical size. The gap between the width and depth of the ablation texture was less than 10 µm, and the gap between the ablation texture spacing was less than 20 μm, which indicated that the ablation texture size agreed well with the theoretical size.

In addition, in order to study the effect of the laser ablation texture on the grinding process, Vickers hardness tester (Jinan, China, Jinan Huayin Testing Instrument Co.) was used to measure the hardness of the tungsten alloy before and after laser ablation. After the tungsten alloy workpiece without texture was ground and polished, and the loading force was set as 1000 gf, 2000 gf and 3000 gf, an indentation was formed on the tungsten alloy surface through a regular diamond indenter, as shown in Figure 4a. Then, the two diagonal lengths of the indentation could be directly measured in the Vickers hardness tester and input into its calculation software. The average hardness of the unabated tungsten alloy was obtained as 452 HV. Then, the nanosecond laser ablation was performed on all of the tungsten alloy surface, based on the parameters in Table 1. The Vickers hardness tester was also used to measure the hardness of the tungsten alloy before and after laser ablation; EDS components of the tungsten alloy surface after ablation were obtained by scanning electron microscopy (Tokyo, Japan, JEOL). Because the surface integrity of the laser ablation tungsten alloy was poor after 1000 gf, 2000 gf and 3000 gf indentation, the ultra-depth field microscope was used to measure the indentation diagonal length, as shown in Figure 4b. Then, Vickers hardness Equation (1) was used to calculate the hardness value of the ablation tungsten alloy [24], and the surface hardness of laser ablation tungsten alloy was 402 HV.
*H* = *KP*/*d*^2^(1)
where *K* is the dimensionless constant, whose value is related to the diamond indenter; *P* is the value of load; *d* is the average of the diagonal length of the indentation.

Figure 5 shows the workpiece morphology and EDS composition results obtained by the scanning electron microscopy (Tokyo, Japan, JEOL) after the nanosecond laser ablation of tungsten alloy. It can be observed from Figure 5a that the bright pure tungsten particles and blended recondensing group with the tungsten phase and bonding phase were separated out on the surface of the tungsten alloy after ablation. As shown in Figure 5b, the content of tungsten in the pure tungsten particle area accounted for 91.03%, which indicated that the combination of tungsten phase and bonding phase was destroyed at a high temperature, so that the tungsten particles with a clear and spherical shape after recondensation were exposed on the surface after ablation. At the same time, the tungsten alloy surface after laser ablation could be observed with the tungsten caking blends’ recondensation; this was because the tungsten alloy material was melted and vaporized in ablating, part of the materials was blown out or evaporated. Another part of the material could not be melted and regelated to form a mass of the recondensation material. As shown in Figure 5c, the main components of the tungsten phase and bond blend recondensation include Tungsten, Nickel, Fe and O, and the content of O is 16.25%, which indicated that redox reactions exist between the metal elements in tungsten alloy and oxygen in high temperature. The microstructure of the tungsten alloy after the nanosecond laser ablation was looser and easier to be removed.

Therefore, after the nanosecond laser ablation, the surface hardness of tungsten alloy was reduced, the tight structure of tungsten alloy was damaged and it was changed as loose structure; moreover, there was a gap between the adjacent recondensation materials. It could be speculated that the surface of tungsten alloy is softened after the laser ablation texture, which would reduce the energy loss in the processing process, and the material would be relatively easy to be removed. It is possible to obtain better surface quality, less subsurface damage, and less grinding force.

### 3.2. Grinding Experiments

The grinding process was carried out on the MGK7120 × 6/F high-precision CNC horizontal spindle rectangular table surface grinder (Wenzhou, China, Wenzhou Wantong Mechanical and Electrical Equipment Co.), as shown in Figure 6, in which the diameter of resin-bonded diamond grinding wheel was 200 mm, the thickness was 20 mm, the wheel granularity was 200#, and the tungsten steel grinding fluid was used. The above tungsten alloy workpieces without ablation and four texture types were used to be ground. In the grinding process, the tungsten alloy workpiece was fixed on the clamping fixture, the clamping fixture was fixed on the dynamometer, and the dynamometer was adsorbed to the electromagnetic force of the surface grinding machine. According to the literature and the grinding experimental experience [7,8,25], the grinding parameters, as shown in Table 3, were designed. Among them, the group 1, 2, 5, 6, 9, 10, 13, 14, 17 and 18 experiments were used to obtain the surface comparison test results, and groups 3, 4, 7, 8, 11, 12, 15, 16 were used to obtain the force signal data with different parameters. The groups used for measuring the force were under the conditions of 3000 r/min wheel speed and 7 μm grinding depth, and the feed rate was 2 m/min, 4 m/min, 6 m/min and 8 m/min. Other groups used for measuring force were under the condition of the 3000 r/min wheel speed and 6 m/min feed rat, and the grinding depth was set as 2 µm, 7 µm, 12 µm and 17 µm, respectively. The parameters of each group were measured twice and the average value was taken. After grinding, the surface roughness of the tungsten alloy was detected by the JITAI820 surface roughness tester, and the surface morphology and EDS components of tungsten alloy were detected by the JSM-IT100 scanning electron microscope (Tokyo, Japan, JEOL).

## 4. Comparative Analysis of Grinding Results

### 4.1. Surface Topography

Figure 7 showed the surface morphology of each workpiece (non-textured surface, texture A, texture B, texture C, texture D) after grinding with the same grinding parameters (the grinding wheel speed was 3000 r/min, feed rate was 6mm/min and grinding depth was 17 μm). As shown in Figure 7, there were grooves, crushing, tungsten microcracks, material rolling, mixing and blending between the bonding phase and tungsten phase, adhesion of debris and cracks on each workpiece surface. However, the surface groove depth, groove internal crack, tungsten phase breakage and crack of the non-textured tungsten alloy workpiece were relatively serious after grinding, while the surface groove depth, groove internal crack, tungsten phase breakage and crack on the surfaces of the ablation texture tungsten alloy workpiece were improved. At the same time, the contrast differences in surface damage between textures were not obvious. It should be specially pointed out that the cracks along the direction of the grooves and the radial cracks with a certain angle to the groove direction appeared in the inner and edge of grinding grooves on the tungsten alloy surface without texture, while some shallow radial cracks only appeared at the bottom of the groove on the tungsten alloy surfaces with textures.

### 4.2. Grinding Force

The grinding force could reflect the change of the contact effect between the diamond abrasive and tungsten alloy, which is an important factor affecting the precision and quality of grinding [26,27]. The grinding force mainly comes from the friction between the abrasive particles and the workpiece, the friction between the binder and the grinding wheel, the elastic-plastic deformation of materials and the removal of abrasive particles on materials [28]. In the grinding process of the tungsten alloy, the normal grinding force and tangential grinding force of the non-textured tungsten alloy and four kinds of textured tungsten alloy were detected by dynamometer, and the variation of the grinding force was observed by changing the feed rate of the workpiece and grinding depth of the grinding wheel. Figure 8 shows the variation of the grinding force when the feed rate of 2 m/min, 4 m/min, 6 m/min and 8 m/min workpiece were selected, respectively, while the grinding depth was 7 µm and the grinding wheel speed was 3000 r/min. It can be observed from Figure 8 that the normal force and tangential force both increased with the increasing feed speed, because the increasing feed speed will increase the thickness of undeformed chips and the contact arc length, which would increase the actual grinding depth of the abrasive particles and the effective abrasive number.

Figure 9 shows the variation of grinding force when grinding depths of 2 µm, 7 µm, 12 µm and 17 µm were selected, respectively, while the workpiece feed rate was 6 m/min and the grinding wheel speed was 3000 r/min. It can be observed from the figure that the normal force and tangential force increase with the increasing grinding depth, because the maximum undeformed chip thickness and contact arc length increase with the increasing grinding depth, which would increase the cutting depth and effective abrasive number of abrasive particles.

As shown in Figure 8 and Figure 9, with the same processing parameters, compared with the non-textured tungsten alloy workpiece, the normal force and tangential force when grinding tungsten alloy workpiece with textures were lower, and the grinding force of texture A was at the minimum; meanwhile, the grinding forces of other textured workpieces were a little higher than texture A, and the grinding forces of the other three textures were similar. The experimental results confirmed that the grinding force was lower when grinding tungsten alloy workpieces with textures to a certain extent, and the tungsten alloy material was easy to be removed. The maximum value of the normal grinding force and tangential grinding force difference between the grinding texture A and non-textured tungsten alloy were 17.54 N and 5.4 N, respectively, when the grinding wheel speed was 3000 rpm and grinding depth was 7 μm; the grinding speed was set to 2 m/min. Meantime, the maximum value of the normal grinding force and tangential grinding force difference between grinding texture A and the non-textured tungsten alloy were 19.15 N and 5.8 N, respectively, when the grinding speed was set to 4 m/min.In addition, the maximum value of the normal grinding force and tangential grinding force difference between grinding texture A and non-textured tungsten alloy were 22.19 N and 6.2 N, respectively, when the grinding speed was set to 6m/min. Moreover, the maximum value of the normal grinding force and the tangential grinding force difference between grinding texture A and the no-textured tungsten alloy were 29.34 N and 6.63 N, respectively, when the grinding speed was set to 8 m/min. From the data on the grinding force difference between the non-textured and texture A, it can be concluded that the ablation textures can reduce the grinding force more effectively with the increasing feed speed.

### 4.3. Surface Roughness

Figure 10 shows the comparison of the ground surface roughness of tungsten alloy with and without texture. As shown in Figure 10, with the increasing grinding depth, the surface roughness of the tungsten alloy increased. The reason is that the removal amount of material increases with the increasing grinding depth, the grinding force increased, and its acting stress on the material increased, resulting in the reduction of grinding quality. It can be observed from Figure 10 that the surface roughness of tungsten alloy with textures was better than that of tungsten alloy without texture, and the surface roughness decreased by 0.023 µm–0.204 µm. The reason why the surface roughness of tungsten alloy with textures was improved after grinding is that the maximum chip thickness of the tungsten alloy with textures is smaller than that of ordinary grinding [19]. Among the four textures, the smallest roughness was found on texture A after grinding, which is corresponding to the minimum grinding force of texture A in the grinding force comparison. According to the design size of textures, the ablation area of texture A can be calculated as 49.255 mm^2^, which is the largest among the four textures, and is also one of the reasons for the excellent performance of texture A in the four textures. Meantime, for similar area ratios of the other textures that were ablated on the workpieces, the roughness was similar and close.

### 4.4. Superiority Analysis of Texture-Assisted Grinding

In summary, when grinding the tungsten alloy assisted by laser ablation textures, the ground surfaces of the tungsten phase breakage, groove crack, groove depth and other conditions could be improved, and the grinding force and surface roughness was reduced; moreover, texture A was more effective than the other textures in improving the grinding quality of the tungsten alloy. This was mainly caused by three reasons. On the one hand, the texture groove of the laser ablation is beneficial for storing grinding debris, facilitating chip removal, storing grinding fluid and avoiding the grinding wheel clog, which makes the extrusion friction of the tungsten alloy workpiece less; the temperature is lower and the grinding force is lower [29,30]. On the other hand, the hardness of tungsten alloy is reduced after laser ablation, and the tungsten alloy material is softened, which makes the tungsten alloy workpiece with the micro texture easier to be removed, so that the grinding force is reduced and the surface quality is better. Finally, for the texture that makes tungsten alloy surface in a discrete state, when extruding and rubbing between the grinding wheel and workpiece, the discrete material is more easily removed because of its lacking constraint. Meanwhile, the grinding effect is related to the area ratio of the texture, for the area ratio of texture A was the most, and the grinding effect with texture A was more outstanding in improving the quality of the tungsten alloy.

## 5. Conclusions

By conducting ordinary grinding and laser ablation texture auxiliary grinding tungsten alloy experiments, the effects of the grinding different tungsten alloy workpieces were contrasted and analyzed, including the ground surface morphology, grinding force, and surface roughness. Moreover, the advantages of the laser ablation texture-assisted grinding tungsten alloy were revealed. The main conclusions are as follows:(1)Grooves, crushing, tungsten microcracks, material roll, the mixing and blending of the bonding phase and tungsten phase, and the adhesion of chipping could be found on the surface morphology of the tungsten alloy workpiece after both ordinary grinding and laser ablation texture-assisted grinding. However, the surface morphology of the laser ablation texture-assisted grinding is better than that of ordinary grinding.(2)Compared with ordinary grinding, the grinding force was reduced effectively with texture A on tungsten alloy, the normal force decreased by 49.91–59.46% and the tangential force decreased by 44.11–58.49%. The surface roughness could be reduced with the laser ablation texture-assisted grinding under the same working condition, the surface roughness decreased by 0.023 µm–0.204 µm and the best improvement ability of the surface roughness among the four types of textures was observed with texture A on the tungsten alloy.(3)There are three main reasons why laser ablation texture-assisted grinding can improve grinding quality and effectively reduce grinding force. On the one hand, the texture groove of the laser ablation is beneficial for storing grinding debris, facilitating chip removal, storing grinding fluid and avoiding grinding the wheel clog, which makes the extrusion friction of tungsten alloy workpiece less, the temperature lower and the grinding force lower. On the other hand, the hardness of the tungsten alloy is reduced after laser ablation, and the tungsten alloy material is softened, which makes the tungsten alloy workpiece with micro texture easier to be removed, so that the grinding force is reduced and the surface quality is better. Finally, for the texture making the tungsten alloy surface in a discrete state, when extruding and rubbing between the grinding wheel and workpiece, the discrete material is more easily removed because of lacking constraint.(4)Finally, the grinding effect is related to the area ratio of the texture. For the area ratio of texture A being the most, the grinding effect with texture A was more outstanding in improving the quality of the tungsten alloy. Meanwhile, for similar area ratios of the other textures ablated on the workpieces, the grinding forces and roughness were similar and close.

## Figures and Tables

**Figure 1 materials-15-07028-f001:**
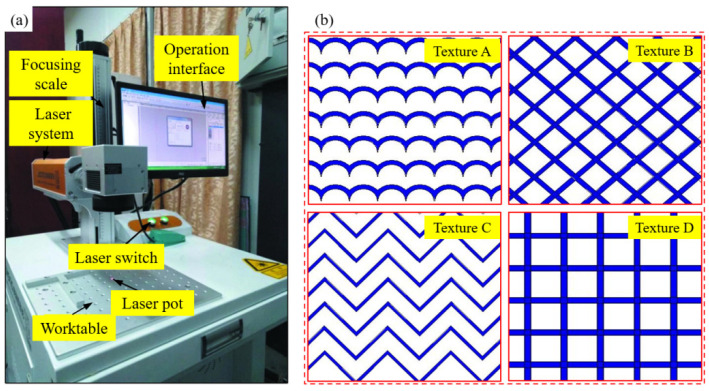
Laser ablation setup and textures mode (**a**) laser ablation system; (**b**) the designed ablation textures.

**Figure 2 materials-15-07028-f002:**
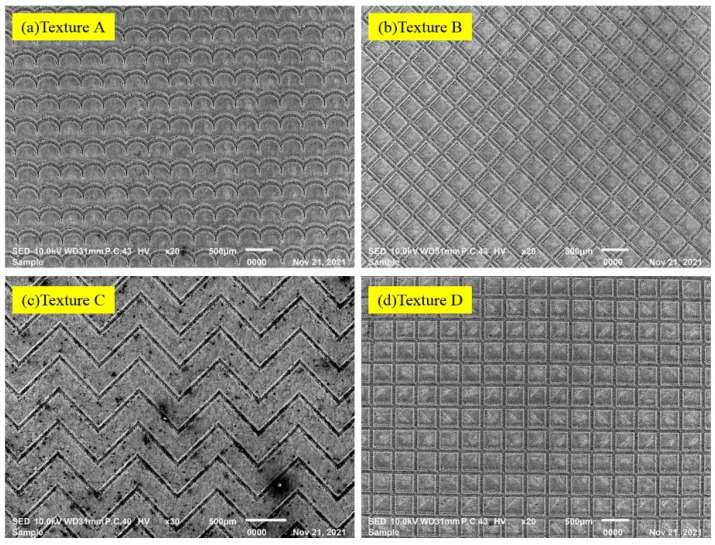
SEM morphology of laser ablation textures (**a**) texture A, (**b**) texture B, (**c**) texture C and (**d**) texture D.

**Figure 3 materials-15-07028-f003:**
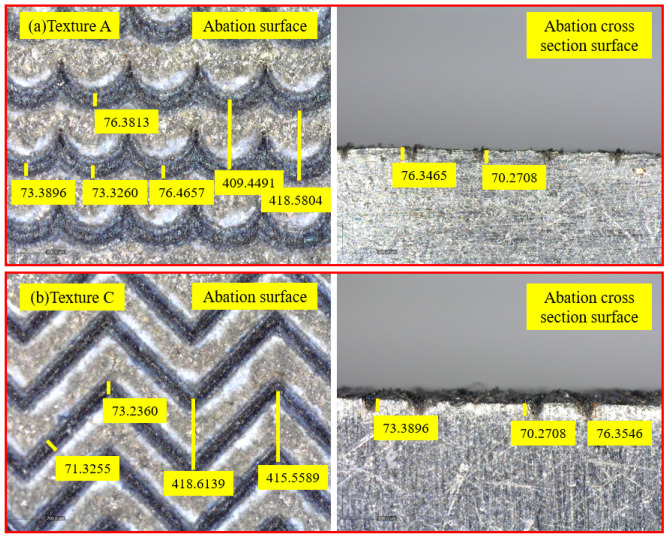
Actual size of laser textures observed by electron microscopy (**a**) texture A; (**b**) texture C.

**Figure 4 materials-15-07028-f004:**
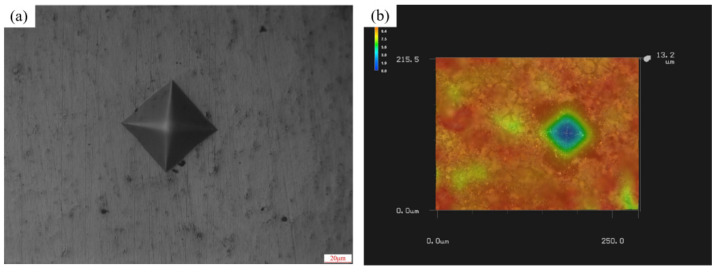
Tungsten alloy indentation; (**a**) indentation morphology before ablation; (**b**) indentation morphology after ablation.

**Figure 5 materials-15-07028-f005:**
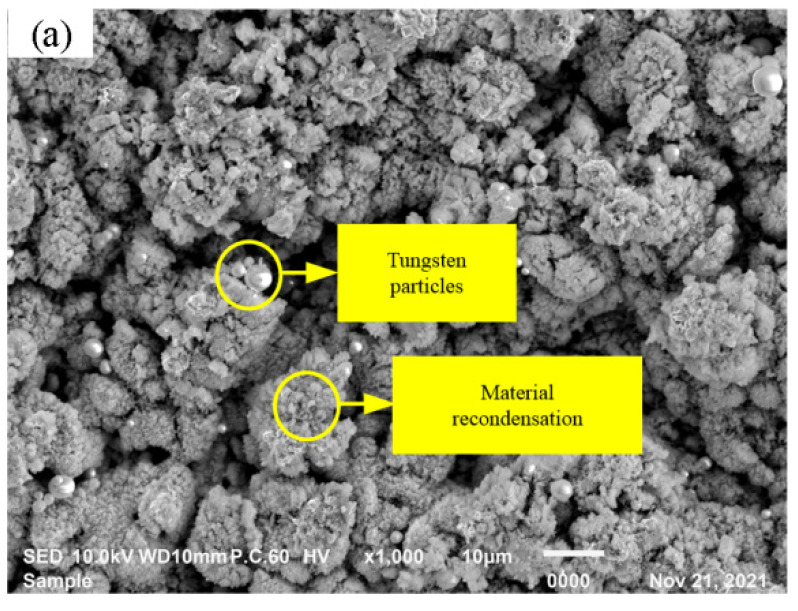
EDS inspection of laser ablation tungsten alloy at different positions; (**a**) SEM morphology of laser ablation tungsten alloy surface; (**b**) EDS of pure tungsten position; (**c**) blended recondensing position.

**Figure 6 materials-15-07028-f006:**
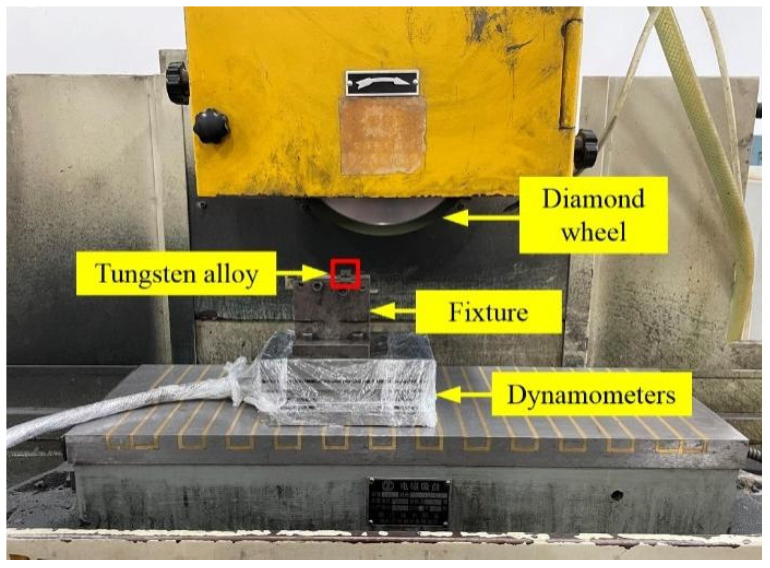
Surface grinding setup.

**Figure 7 materials-15-07028-f007:**
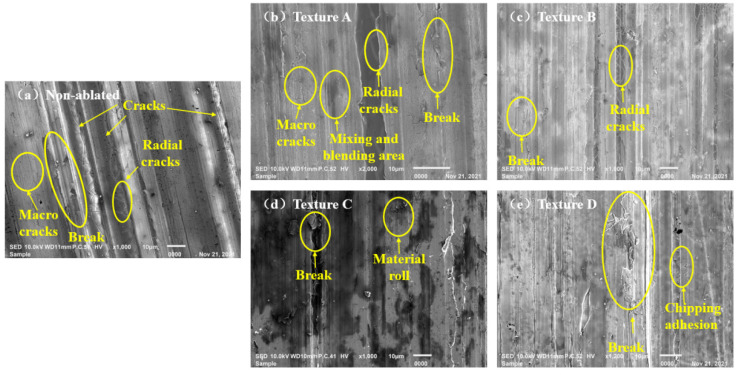
SEM morphology of different texture surfaces after grinding; (**a**) non-ablated workpiece; (**b**) texture A; (**c**) texture B; (**d**) texture C; (**e**) texture D.

**Figure 8 materials-15-07028-f008:**
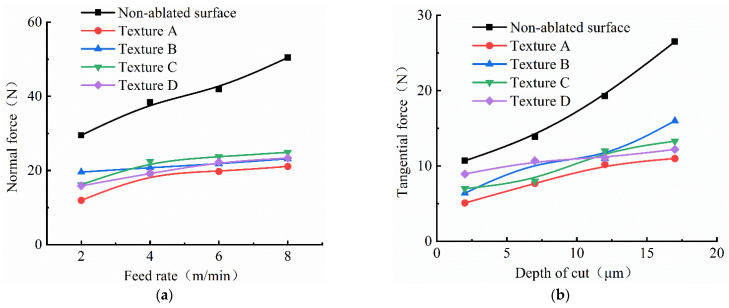
Influence of feed rate on grinding force with different textures; (**a**) feed rate vs. normal force; (**b**) feed rate vs. normal force.

**Figure 9 materials-15-07028-f009:**
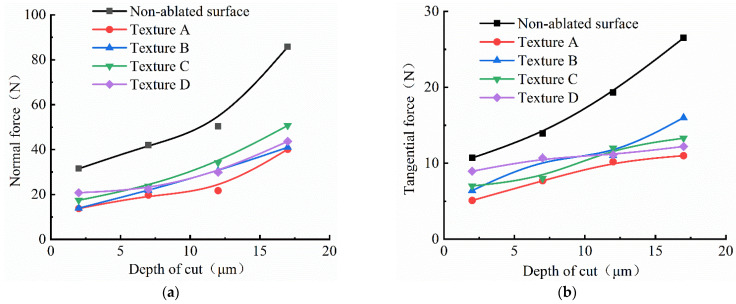
Influence of depth of cut on grinding force with different textures; (**a**) depth of cut vs. normal force; (**b**) depth of cut vs. normal force.

**Figure 10 materials-15-07028-f010:**
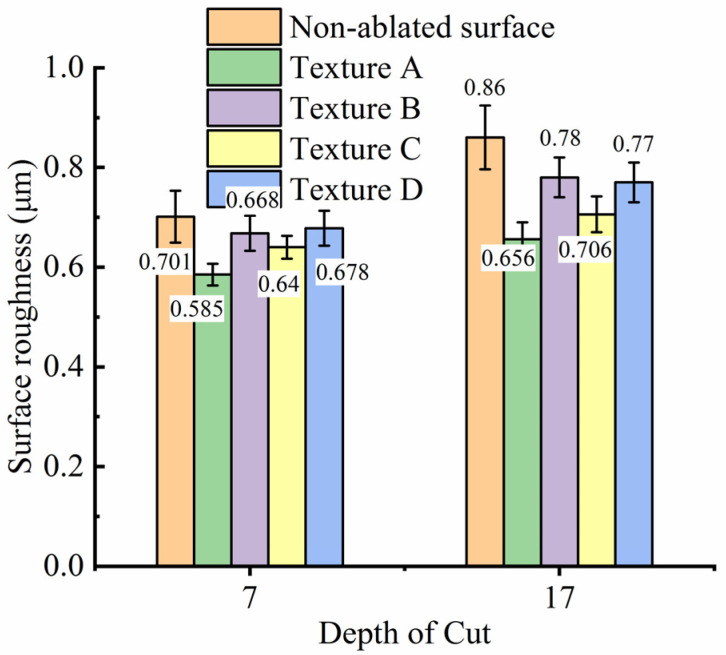
Surface roughness with different textures.

**Table 1 materials-15-07028-t001:** Components of tungsten alloy.

Alloy Designation	Content of Alloying Element (Mass Fraction /%)			Content of Impurity (Mass Fraction /%) ≤			
	W	Ni	Fe	Al	Mg	C	O
95W-3.5Ni-1.5Fe	About 95	3.4–3.6	1.4–1.6	0.002	0.003	0.008	0.005

**Table 2 materials-15-07028-t002:** Nanosecond laser ablation parameters.

Parameters	Power/(W)	Scanning Speed/(mm/s)	Pulse Frequency/(kHz)	Pulse Width/(ns)
Value	20	100	50	20

**Table 3 materials-15-07028-t003:** Grinding parameters.

No.	Grinding Wheel Speed (r/min)	Feed Rate (m/min)	Depth of Cut (μm)	Texture Mode	Note
1	3000	6	2	Without texture	
2	3000	6	17	Without texture	
3	3000	2/4/6/8	7	Without texture	Only used for measuring force
4	3000	6	2/7/12/17	Without texture	Only used for measuring force
5	3000	6	7	Texture A	
6	3000	6	17	Texture A	
7	3000	2/4/6/8	7	Texture A	Only used for measuring force
8	3000	6	2/7/12/17	Texture A	Only used for measuring force
9	3000	6	7	Texture B	
10	3000	6	17	Texture B	
11	3000	2/4/6/8	7	Texture B	Only used for measuring force
12	3000	6	2/7/12/17	Texture B	Only used for measuring force
13	3000	6	7	Texture C	
14	3000	6	17	Texture C	
15	3000	2/4/6/8	7	Texture C	Only used for measuring force
16	3000	6	2/7/12/17	Texture C	Only used for measuring force
17	3000	6	7	Texture D	
18	3000	6	17	Texture D	
19	3000	2/4/6/8	7	Texture D	Only used for measuring force
20	3000	6	2/7/12/17	Texture D	Only used for measuring force

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
