# Peer review of "Experimental Study on Laser Ablation Texture-Assisted Grinding of Tungsten Alloy"

_materials, 2022, doi:10.3390/ma15197028_

Round 1
Reviewer 1 Report
This work describes the treatment of the surface of tungsten alloy samples by laser ablation texture and precision grinding. The paper presents many problems and is not well written and organized. It could be accepted for publication only after several changes.
1) The English should be completely revised since many sentences are really difficult to understand.
2) The paper is not well organized. The experimental section contains al lot of material which could be reported in the results. The division of the sections is confusing, and it does not facilitate the understanding. For example, why is “Superiority analysis of texture-assisted grinding” considered a sub-section of “Surface roughness”?
3) The authors use different types of laser textures, but they do not discuss if the different textures play a role in the final results.
4) In the experimental, the authors state: “After ablation, the surface morphology of four textures was observed by electron microscopy and scanning electron microscopy (JSM-IT100)”. SEM images were reported in the paper but what about the other “electron microscopy”?
5) What is the reason of the presence of Strontium in the EDS spectra?
6) Table 1 reports Chinese symbols.
7) Table 2 is incomplete. Repetition, followed by a number, means nothing for a laser source. Probably the authors meant “Repetition rate” but that case the units must be reported (Hz, kHz?). In the same way, I imagine that the pulse width is in nm!
8) In General, the figure captions are not descriptive enough.
9) SEM images of Figure 7 are useless since they are very similar.
10) Figure 10 is not clear. Anyway, it does not show the surface roughness.
Author Response
Dear editor(s) and reviewer(s),
On behalf of my co-authors, we appreciate you and reviewers very much for the constructive comments and suggestions on our manuscript entitled “Experimental Study on Laser Ablation Texture Assisted Grinding of Tungsten Alloy(No. 1944763)”.
We have revised the manuscript according to the comments raised by the reviewers and editor with the amendments in red font correspondingly. The responses to the reviewers’ comments are listed below.
Thank you and best regards.
Yours sincerely,
Bing Chen
Reviewer1' Comments to Author:
Comments and Suggestions for Authors
This work describes the treatment of the surface of tungsten alloy samples by laser ablation texture and precision grinding. The paper presents many problems and is not well written and organized. It could be accepted for publication only after several changes.
(1) The English should be completely revised since many sentencesare really difficult to understand.
Response: Thanks for the reviewer’s suggestion. According to this comment, the English in this paper has been checked and revised carefully.
(2) The paper is not well organized. The experimental section contains a lot of material which could be reported in the results. The division of the sections is confusing, and it does not facilitate the understanding. For example, why is “Superiority analysis of texture-assisted grinding”considered a sub-section of “Surface roughness”?
Response: Thanks for the reviewer’s suggestion. According to this comment, part of the manuscript sections has been adjusted and organized, and the division of the section “Superiority analysis of texture-assisted grinding” in the rough manuscript was a mistake, it has been revised.
“1 Components of experimental tungsten alloy”
“3.3 Surface roughness”
“3.4 Superiority analysis of texture-assisted grinding”
(3) The authors use different types of laser textures, but they do not discuss if the different textures play a role in the final results.
Response: Thanks for the reviewer’s suggestion. According to this comment, the different textures play a role in the final results was discussed.
“(4) Finally, the grinding effect is related to the area ratio of texture. For the area ratio of texture A was the most, the grinding effect with texture A was more outstanding in improving the quality of the tungsten alloy. Meantime, for similar area ratios of the other textures was ablated on the workpieces, the grinding forces and roughness was similar and close.”
(4) In the experimental, the authors state: “After ablation, the surface morphology of four textures was observed by electron microscopy and scanning electron microscopy (JSM-IT100)”. SEM images were reported in the paper but what about the other “electron microscopy”?
Response: Thanks for the reviewer’s suggestion. According to this comment, the Figure 3 was tested by electron microscopy. And it was described in the manuscript as:
“As shown in Figure 3 which observed by electron microscopy, ...”
Figure 3 Actual size of laser textures observed by electron microscopy (a) texture A (b) texture C
(5) What is the reason of the presence of Strontium in the EDS spectra?
Response: Thanks for the reviewer’s suggestion. Strontium should not be present in the Fig.3, in fact, tiny amounts of Aluminum and Magnesium should be oxidized and observed in the test data. According to this comment, the data of Figure 3 was recalculated and revised based on the previous EDS data.
Figure 5 EDS inspection of laser ablation tungsten alloy at different positions (a) SEM morphology of laser ablation tungsten alloy surface (b) EDS of pure tungsten position (c)blended recondensing position
(6) Table 1 reports Chinese symbols.
Response: Thanks for the reviewer’s suggestion. According to this comment, the Table 1 has been revised as:
Table 1 Compositions of tungsten alloy
|
Alloy designation |
Content of alloying element (mass fraction /%) |
Content of impurity (mass fraction /%) ≤ |
|||||
|
W |
Ni |
Fe |
Al |
Mg |
C |
O |
|
|
95W-3.5Ni-1.5Fe |
About 95 |
3.4-3.6 |
1.4-1.6 |
0.002 |
0.003 |
0.008 |
0.005 |
(7) Table 2 is incomplete. Repetition, followed by a number, means nothing for a laser source. Probably the authors meant “Repetition rate” but that case the units must be reported (Hz, kHz?). In the same way, I imagine that the pulse width is in nm!
Response: Thanks for the reviewer’s suggestion, the repetition frequency was expressed as emission times of laser pulse in 1 second, according to this comment, the “Repetition frequency ” has been revised as “Pulse frequency”in Table 2. And, the “Pulse width” was used to express the emission time of one laser pulse, thus the unit of pulse width should be “ns”.
Table 2 Nanosecond laser ablation parameters
|
Parameters |
Power/(W) |
Scanning speed/(mm/s) |
Pulse frequency/(kHz) |
Pulse width/(ns) |
|
Value |
20 |
100 |
50 |
20 |
(8) In General, the figure captions are not descriptive enough.
Response: Thanks for the reviewer’s suggestion, according to this comment, the figure captions has been revised as:
“Figure 1 Laser ablation setup and textures mode (a) laser ablation system, (b) the designed ablation textures”
“Figure 2 SEM morphology of laser ablation textures (a) texture A (b) texture B (c) texture C (d) texture D”
“Figure 3 Actual size of laser textures observed by electron microscopy (a) texture A (b) texture C”
“Figure 5 EDS inspection of laser ablation tungsten alloy at different positions (a) SEM morphology of laser ablation tungsten alloy surface (b) EDS of pure tungsten position (c)blended recondensing position”
“Figure 6 Surface grinding setup”
“Figure 7 SEM morphology of different texture surfaces after grinding (a)non-ablated workpiece(b) texture A (c) texture B (d) texture C (e) texture D”
“Figure 8 Influence of feed rate on grinding force with different textures (a) feed rate VS normal force (b) feed rate VS normal force ”
“Figure 9 Influence of depth of cut on grinding force with different textures (a) depth of cut VS normal force (b) depth of cut VS normal force”
“Figure 10 Surface roughness with different textures”
(9) SEM images of Figure 7 are useless since they are very similar.
Response: Thanks for the reviewer’s suggestion, though there little differences between different SEM images, the Figure 7 could be used to show the similar characteristics and the actual surfaces. And the cracks along the direction of the grooves and the radial cracks with a certain angle to the groove direction, which appeared in the inner and edge of grinding grooves on the tungsten alloy surface without texture, while some shallow radial cracks only appeared at the bottom of the groove on the tungsten alloy surfaces with textures.
(10) Figure 10 is not clear. Anyway, it does not show the surface roughness.
Response: Thanks for the reviewer’s suggestion. According to this comment, the Figure 10 has been revised as:
Figure 10 Surface roughness with different textures

Reviewer 2 Report
The manuscript has major issues, I do have concerns:
1. Please explain the novelty of the current work. The reviewer is not able to understand how this paper adds value to the scientific literature.
2. Use recent literature such as https://www.mdpi.com/1996-1944/11/5/801/htm , https://asmedigitalcollection.asme.org/tribology/article/127/1/248/462965/State-of-the-Art-in-Laser-Surface-Texturing, https://www.sciencedirect.com/science/article/pii/S0924013604009707, https://www.sciencedirect.com/science/article/pii/S003039922200367X .
3. Figure 10 - Add error bars (standard deviation)
4. Table 3: On what basis were the grinding parameters selected.
5. What is the use of grinding a textured surface?
6. Mention the hardness values in whole numbers.
Author Response
Dear editor(s) and reviewer(s),
On behalf of my co-authors, we appreciate you and reviewers very much for the constructive comments and suggestions on our manuscript entitled “Experimental Study on Laser Ablation Texture Assisted Grinding of Tungsten Alloy(No. 1944763)”.
We have revised the manuscript according to the comments raised by the reviewers and editor with the amendments in red font correspondingly. The responses to the reviewers’ comments are listed below.
Thank you and best regards.
Yours sincerely,
Bing Chen
Reviewer2' Comments to Author:
Comments and Suggestions for Authors
The manuscript has major issues, I do have concerns:
(1) Please explain the novelty of the current work. The reviewer is not able to understand how this paper adds value to the scientific literature.
Response: Thanks for the reviewer’s suggestion. According to this comment, the scientific literature of this paper was proposed in the introduction.
“At present, there are a lot of studies on cutting, grinding and polishing of tungsten alloys, however, there are many problems in traditional cutting and grinding, meantime, the laser ablation texture grinding technology has been used to machine other diffcult-to-cut materials with high efficiency and surface qulity, while there are few reports on high efficiency, precision grinding or laser ablation texture assisted grinding of tungsten alloy. Therefore, experimental research on laser ablation texture assisted grinding of tungsten alloy was carried out in this paper. Through comparing of the surface morphology, grinding force and surface roughness of tungsten alloy machined by ordinary grinding and laser ablation texture assisted grinding, the advantages of laser ablation texture assisted grinding for tungsten alloy were discussed. This paper would provide guidance for realizing high efficiency and precision machining of tungsten alloy.”
(1) Use recent literature such as https://www.mdpi.com/1996-1944/11/5/801/htm , https://asmedigitalcollection.asme.org/tribology/article/127/1/248/462965/State-of-the-Art-in-Laser-Surface-Texturing, https://www.sciencedirect.com/science/article/pii/S0924013604009707, https://www.sciencedirect.com/science/article/pii/S003039922200367X .
Response: Thanks for the reviewer’s suggestion. According to this comment, these recent literature has been cited in the manuscript.
[11] Bonse, J.; Kirner, S.V.; Griepentrog, M.; Spaltmann, D.; Krüger, J. Femtosecond laser texturing of surfaces for tribological applications. MATERIALS. 2018, 11(5), 801.
[13] Santosh, S.; Thomas, J.K.; Pavithran, M.; Nithyanandh, G.; Ashwathet, J. An experimental analysis on the influence of CO2 laser machining parameters on a copper-based shape memory alloy. Opt. Laser. Technol. 2022, 153, 108210.
[14] Etsion, I. State of the art in laser surface texturing. J. Trib. 2005, 127(1), 248-253.
[20] Du, D.; He, Y.F.; Sui, B.; Xiong, L.J.; Zhang, H. Laser texturing of rollers by pulsed Nd: YAG laser. J. Mater. Process. Tech. 2005, 161(3), 456-461.
(3) Figure 10 - Add error bars (standard deviation)
Response: Thanks for the reviewer’s suggestion. According to this comment, the Figure 10 has been revised as:
Figure 10 Surface roughness with different textures
(4) Table 3: On what basis were the grinding parameters selected.
Response: Thanks for the reviewer’s suggestion. The grinding experiments of tungsten alloy has been carried out in our previous research, much experience has been accumulated. Therefore, the grinding parameters in this manuscript selected according to the literature and the previous grinding experimental experience.
(5) What is the use of grinding a textured surface?
Response: Thanks for the reviewer’s suggestion. The textured surfaces could help the tungsten alloy be removed efficiently with high surface quality and low grinding force. And it was concluded in the conclusion.
(6) Mention the hardness values in whole numbers.
Response: Thanks for the reviewer’s suggestion. According to this comment, the hardness values has been revised in whole numbers.
“The average hardness of the unabated tungsten alloy was obtained as 452HV. And then, nanosecond laser ablation was performed on all the tungsten alloy surface based on the parameters in Table 1. Vickers hardness tester was also used to measure the hardness of the tungsten alloy before and after laser ablation, EDS components of the tungsten alloy surface after ablation were obtained by scanning electron microscopy. Because the surface integrity of laser ablation tungsten alloy was poor after 1000 gf, 2000 gf, 3000 gf indentation, ultra depth field microscope was used to measure the indentation diagonal length, as shown in Figure 4 (b). And then, the vickers hardness formula (1) was used to calculate the hardness value of the ablation tungsten alloy, and the surface hardness of laser ablation tungsten alloy is 402HV.”

Round 2
Reviewer 1 Report
In the present form the paper can be accepted for publication. Indeed the paper is well organized, the English is satisfactory, the references are adequate and, finally, the topic is of interest for the community.
Author Response
Thanks for the reviewer’s suggestion. The English in this manuscript has been checked and revised carefully.
Reviewer 2 Report
Following comments are still not addressed in the revised paper.
(3) Figure 10 - Add error bars (standard deviation). Not yet done.
(4) Table 3: On what basis were the grinding parameters selected. Authors cant select on previous experience. They should have some scientific method to select the parameters.
Author Response
Thanks for the reviewer’s suggestion. The English in this manuscript has been checked and revised carefully.
Point 1: Figure 10 - Add error bars (standard deviation). Not yet done.
Response 1: Thanks for the reviewer’s suggestion. The error bars has been added in Figure 10.
Point 2: Table 3: On what basis were the grinding parameters selected. Authors cant select on previous experience. They should have some scientific method to select the parameters.
Response 2: Thanks for the reviewer’s suggestion. Indeed, the selection of grinding parameters should be based on some scientific method. However, in grinding process, the selection of grinding parameters depends on many factors, such as the stability and accuracy of grinder, the granularity and bond type of grinding wheel, material characteristics, grinding method and so on. In the experiments of this manuscript, for the grinder, grinding wheel, grinding method has been confirmed, the basic of grinding parameters selection could be new research topic, but the basic of grinding parameters selection is not the core research content of this manuscript. Therefore, the grinding parameters in the manuscript selected according to the literature, experimental conditions and the previous grinding experimental experience.
The selection experience of grinding parameters could be concluded from three references as follow:
[7]Zhou, Z.Z. Research on Electrolytic Grinding Process of Tungsten Alloy. M.S.Thesis, Dalian University of Technology, May 2019.
Tab.5.1 Factors— level table
|
Project |
Factor A (rpm) |
Factor B (mm/s) |
Factor C (μm) |
|
Level 1 |
6000 |
0.5 |
30 |
|
Level 2 |
8000 |
1.5 |
50 |
|
Level 3 |
10000 |
2.5 |
70 |
[8] Xu, L. Wear Mechanism Of Diamondtool In Rotary Ultrasonic Grinding Of Tungsten Alloy. M. S. Thesis, Southwest Jiaotong University, May 2021.
Tab.5.1 Test parameters for tungsten alloy grinding
|
Variable |
Numerical value |
|
Workpiece material and dimension (mm) |
95W(φ60mm×5mm) |
|
Grinding conditions |
dry grinding |
|
Feeding speed (mm/min) |
20 |
|
Spindle speed (rpm) |
5000 |
|
Grinding depth (μm) |
10,15,20 |
|
Ultrasonic amplitude (μm) |
0.2 |
[30] Li, J.Y.; Jin, Z.J. Experimental study on Grinding Process of High-purity Tungsten. Aeronautical Manufacturing Technology(in Chiniese). 2017,06,55-59.
Tab.2 Grinding wheel comparison grinding test parameters
|
Grinding test conditions |
Numerical value |
|
Grinding wheel line speed /(m·s-1) |
23 |
|
Grinding depth/μm |
6,12 |
|
Workbench speed /(m·min-1) |
12 |
|
Grinding fluid |
Water-based emulsion |
Based on these three references, the linear velocity of grinding wheel could be selected from 20m/s to 40m/s, the depth of cut could be selected 2μm to 70μm, and the feed rate could be selected from 2m/min to 12 m/min. Therefore, in the experiment of this manuscript, considering our experimental conditions and our previous grinding experimental experience, the linear velocity of grinding wheel was selected from 31.4m/s (3000rpm of wheel speed), the depth of cut was selected 2μm to 17μm, and the feed rate was selected from 2m/min to 8 m/min.